# Effect of Solution-to-Binder Ratio and Alkalinity on Setting and Early-Age Properties of Alkali-Activated Slag-Fly Ash Binders

**DOI:** 10.3390/ma16010373

**Published:** 2022-12-30

**Authors:** Ali Naqi, Brice Delsaute, Markus Königsberger, Stéphanie Staquet

**Affiliations:** 1BATir Department, Université Libre de Bruxelles, CP194/02, 50 Avenue F.D. Roosevelt, 1050 Brussels, Belgium; 2Institute for Mechanics of Materials and Structures, TU Wien, Karlsplatz 13/202, 1040 Vienna, Austria

**Keywords:** alkali-activated materials, setting time, ultrasonic measurements, calorimetry, strength

## Abstract

The growing use of blends of low- and high-calcium solid precursors in combination with different alkaline activators requires simple, efficient, and accurate experimental means to characterize their behavior, particularly during the liquid-to-solid transition (setting) at early material ages. This research investigates slag-fly ash systems mixed at different solution-to-binder (s/b) ratios with sodium silicate/sodium hydroxide-based activator solutions of varying concentrations. Therefore, continuous non-destructive tests—namely ultrasonic pulse velocity (UPV) measurements and isothermal calorimetry tests—are combined with classical slump flow, Vicat, and uniaxial compressive strength tests. The experimental results highlight that high alkali and silica contents and a low s/b ratio benefit the early-age hydration, lead to a faster setting, and improve the early-age strength. The loss of workability, determined from the time when the slump flow becomes negligible, correlates well with ultrasonic P-wave velocity evolutions. This is, however, not the case for Vicat or calorimetry tests.

## 1. Introduction

As an essential construction material, concrete and its main constituent, cement, were widely used in the construction industry for over a century. With rapid infrastructural growth, global cement production has now reached over 4 billion tonnes per annum [1] contributing to 8% of the planet’s total CO_2_ emissions [2]. Alkali-activated materials (AAMs) are promising alternatives to traditional ordinary Portland cement (OPC) systems, as they not only lower CO_2_ emissions but also foster the development of a circular economy by utilizing waste, such as magnesia iron slags [3] or tungsten mine waste [4].

Blast furnace slag and fly ash are the most commonly used solid precursors. They are activated with aqueous hydroxides and/or silicate solutions so that more and more reaction products (hydrates) form and lead to strong and durable alkali-activated materials. Alkaline activation of calcium-rich slag (AAS) forms calcium-aluminosilicate hydrate (C-A-S-H) gel [5,6], which provides high early-age strength [7,8,9] but often leads to non-workable and quick-setting mixes [10]. As for the activation of low-calcium fly ash (AAF), the main hydration product is sodium-aluminosilicate hydrate (N-A-S-H) gel [11], which leads to the desired workability. However, these mixes suffer from low early-age strength [12,13,14], which may be overcome by curing at elevated temperatures [15]. Blending slag and fly ash, together with the choice of suitable activators, overcomes the aforementioned drawbacks and provides somewhat contradicting requirements for modern AAMs: workable mixes and high early-age strengths [16,17,18,19].

Workability is typically assessed by the characterization of the setting behavior of the material. Setting, in turn, refers to the solidification process of a previously liquid material due to a gradual stiffening [20]. The beginning of solidification of the fresh paste, often termed as the “initial set”, refers to the time when the paste loses its plasticity or becomes unworkable [21]. The “final set” refers to the time when the solidification is completed or when measurable mechanical properties start to develop [21,22]. Standardized test methods for the determination of an initial and final set of cement-based materials are based on thresholds for penetration resistance using a standardized test apparatus—see EN 196-3 (for paste) and ASTM C403 (for mortar). These two somewhat arbitrary penetration resistance thresholds do not sufficiently describe the continuous fluid-to-solid transition, particularly not for AAMs, where the setting behavior is often unexpected. 

To better understand and monitor the setting process, continuous and, thus, non-destructive tests (NDT) have been employed, among which the ultrasonic pulse velocity (UPV) experiments are widely used [23,24,25,26,27,28,29,30]. Several suggestions have been made to analyze the measured wave velocities [28,30,31,32,33,34]—see Table 1. Chotard [35] and Smith [31] highlighted two characteristic transition points in the temporal evolution of the P-wave velocity of cement-paste samples and linked them to the setting process. The first transition point (initial set) refers to the time when the P-wave velocity increases significantly and the second transition point signifies a slow increase in velocity on the UPV curve. Trtnik et al. [36] defined the initial set by the first inflection (maximum of the first derivative of the P-wave velocity) for cement-paste mixes. By analogy, Reinhardt et al. [28] successfully correlated the first inflection point of the P-wave velocity to the initiation of the setting and suggested a velocity threshold of 1500 m/s for the final set. De Belie et al. [30] investigated the energy changes of the ultrasound wave on mortar samples and correlated the local maximum of the temporal evolution of the ultrasound energy to the end of workability. Krüger et.al [37] suggested that a dynamic shear modulus of about 0.1 GPa represents a suitable indicator for the initial setting. 

For alkali-activated materials, very few studies are available [38,39,40,41,42,43,44], despite the complex setting behavior and the large variety of material systems. Another frequently used continuous NDT technique is calorimetry testing, i.e., the measurement of the heat emitted during the exothermal chemical reaction of binder and solution. For OPC-based materials, setting occurs in the acceleration stage [27,45,46,47], during which the heat flow increases continuously with time. For AAMs, e.g., for alkali silicate-activated binders, however, setting occurs well before the acceleration period [48]. Herb [49] proposed a bilinear approximation approach, considering UPV as a function of the calorimetry-derived reaction degree to indicate the final setting, while Schindler’s [50] approach included additional consideration of w/c ratios to predict the initial and final setting times. While most of the work on setting identification and mechanical evolution is reported for OPC-based materials, studies on AAMs are very limited: especially considering the influence of precursor nature and activator type on the setting process. A list of criteria, defined in the literature, for initial and final setting determination using different references and experimental methods are summarized in Table 1.

**Table 1 materials-16-00373-t001:** Initial and final setting estimation by different evaluation techniques for OPC systems and AAMs.

Method	Criterion	Mix Scale Initial Set		Final Set
UPV	Velocity threshold (m/s)	Paste	1500 [36,51], 1440–1550 [52] *	1750–1850 [51], 1650–1725 [52] *
		Mortar	800–980 [29]	1200–1400 [29], 1500 [28]
		Concrete	1000–1500 [53], 2300–2700 [49], 1100–2000 [29]	3000 [53], 2790–3180 [49], 2000–3000 [29]
UPV	First inflection point	[28,36,54,55,56] *		-
UPV	Intersection of tangent lines	[31,35,42,57] *		
UPV & Calorimetry	Minimum reaction degree required for setting to occur (%)	Concrete Paste	Schindler [50] prediction = 7.5% Shiva [52] * approximation = 2.1–3.2%	Schindler [50] = 13% Herb [49] bilinear approx. = 8.5% Shiva [52] * = 3.5–4.5%

* Indicate AAMs systems.

In this study, the efficacy of both ultrasonic and calorimetric tests to decipher the setting of alkali-activated slag fly-ash binders was investigated. These continuous NDT techniques were accompanied with classical penetration-resistance (Vicat) tests and slump flow measurements. To monitor the development of mechanical properties of the fragile samples at early ages, close to the setting, uniaxial compressive strength tests were performed with utmost care so as to not damage the samples before testing. This way, characteristic features (such as inflection points) found in the temporal evolution of the ultrasound properties, as well as the heat release, can be directly correlated to changes in the load-bearing capacity of the material. Five different slag-fly ash mixes are compared. The influence of precursors, activator concentration, alkali and silica content, and solution-to-binder ratio (s/b) was thoroughly studied. 

## 2. Raw Materials and Experimental Methods

### 2.1. Raw Materials

Ground granulated blast furnace slag (GGBFS) and class F fly ash (FA) were the main precursors used in this study. The oxide compositions of all the precursor are summarized in Table 2. The Blaine fineness of GGBFS and FA were 516 m^2^/kg and 921 m^2^/kg, respectively. The densities of GGBFS and FA were 2.92 and 2.32 (g/cm^3^), respectively. Sodium hydroxide (NaOH) with 97% purity, in powdered form, and liquid sodium silicate (Na_2_SiO_3_) solution with 28.50% SiO_2_, 18% Na_2_O, and 53.50% H_2_O, by wt., were used as alkaline activators.

### 2.2. Mix Design and Specimen Preparation

The slag-fly ash ratio was kept constant at 50/50 by weight for all the mixes. Alkaline activators (mixture of NaOH solution and liquid sodium silicate) were used to prepare all mixes. The concentration of NaOH solution was 8 M and the modulus of sodium silicate (Ms) value (defined as the mass ratio SiO_2_/Na_2_O) was adjusted to 1.44 by adding NaOH solution. The NaOH solution was prepared a day ahead of casting to allow the heat stemming from the exothermic reaction to dissipate (cooled down to room temperature). The solution (combined activator + additional water) to binder (slag + fly ash) ratio varied from s/b = 0.47 to s/b = 0.70, while water to binder ratio ranged from w/b = 0.36 to w/b = 0.53, as shown in Table 3. The mixtures were designated with initials from the precursors, followed by the s/b ratio (e.g., first labelled mix, SF—050SB, where the S symbol denotes slag, F for fly ash followed by 0.50 s/b). In the case of the last two mixes, SF—047SB—LA, and SF—070SB—HA, the additional labels LA and HA indicate the lower and higher alkali content, respectively.

Mixing was carried out in a Hobart mixer, carefully sticking to the following procedure: The activating solution was poured into the bowl, followed by adding all the precursors. Mixing was carried out for one minute at low-speed (140 ± 5 rpm) and switched to high-speed (185 ± 10 rpm) for another minute. Thereafter, the mix rested for one and a half minutes and, at the same time, any material attached to the mixing paddle, sides, or at the bottom of the bowl was scrapped and added back to the mix. The mixing process ended with one minute of high-speed mixing. The mixture was poured into the molds, as required for the specific test, wrapped in plastic sheets to avoid any evaporation, and placed into a climatic chamber at 20 ± 1 °C and 70% RH until testing.

### 2.3. Experimetnal Methods

#### 2.3.1. Slump Flow

The workability loss of fresh slag-fly ash binders was investigated using a mini-slump flow test [59,60]. Immediately after mixing, the fresh paste was poured into a conical mold with a top opening with an inner diameter of 70 ± 0.5 mm), a height of 50 ± 0.5 mm, and a bottom opening with an inner diameter of 100 ± 0.5 mm, as described in ASTM C230 [61]. The mold was placed in the center of a clean and lubricated plate, filled with fresh paste, and, subsequently, lifted to allow the paste to spread. The average of four spread diameter readings was recorded as soon as the flow stopped. The procedure was repeated at intervals of 10 to 15 min until virtually no spreading was further observed. The test was conducted once for each composition.

#### 2.3.2. Vicat Test

Setting-time characterization was carried out using a classical Vicat needle, in accordance with EN 196–3 [62]. Immediately after mixing, the fresh paste was poured into a lightly oiled conical rubber mold, with a top inner diameter of 70 ± 0.5 mm, a height of 40 ± 0.2 mm, and a bottom inner diameter of 80 ± 0.5 mm. and placed on a base plate. The paste-filled molds were immersed in water controlled at 20 ± 1 °C. The initial setting time was recorded as the time elapsed between the starting time (when the precursors first came in contact with the alkaline activator) and the time when the distance between the penetrating needle and base plate was 6 ± 3 mm. The final setting time was recorded as the time measured from the starting time to the time when the needle was only able to penetrate 0.5 mm into the paste. Vicat tests were performed once for each composition. 

#### 2.3.3. Isothermal Calorimetry

The heat-flow evolution related to the exothermal chemical reaction was continuously monitored using an eight-channel TAM Air isothermal calorimeter. Heat-flow q was measured and translated to cumulative heat Q by integration, reading as:(1)$$Q\;\left(t\right)=\int_{0}^{t}q\left(t\right)\cdot \mathrm{dt}$$

After mixing, 2 samples of about 10 g were collected and directly inserted into the isothermal calorimeter. The test was started at an age of about 15 min for each composition. The heat-flow data were recorded up to ages of 96 h at a fixed container temperature of 20 ± 0.01 °C. Later calorimetry measurements (more than 96 h) were not sufficiently precise, due to decreasing heat flow values. The test was performed once for each composition. 

#### 2.3.4. Ultrasonic Measurements

The measurement of P-wave transmission velocity was carried out using a FreshCON system [28] equipped for monitoring P-wave transmission on cementitious materials, as shown in Figure 1. The system consisted of two polymethacrylate (PMMA) walls with semi-embedded sensors on both external sides of the walls, tied together by four screws and separated by a U-shape rubber foam. An ultrathin polyimide film was glued onto the sensors as protection. The mold thickness for the paste mixtures was 2.5 cm, the width was 10 cm, and the height was 11 cm. Every minute, a high-voltage pulse generator produced a 5 µs pulse that was sent through the mix-filled container by a cylindrical broadband piezo-electric transducer (with 0.5 MHz center frequency). The signals produced by the sensor situated on the other side of the sample (receiver) were sent to the computer through a DAQ (data acquisition) card. The ultrasonic measurements started immediately after casting and were performed continuously until the material aged 72 h. The containers were plastic-wrapped and kept in a climatic chamber at 20 ± 1 °C throughout the entire testing. The sought P-wave velocity $V_{P}$ was computed as: (2)$$V_{P}=\frac{D}{t-d}$$
where D is the distance between the sensors, t is the picked time at signal onset and d is the delay time defined by the container. Testing was carried out once for each mix. 

#### 2.3.5. Compressive Strength

The compressive strength of slag-fly ash paste samples was tested on 50 mm large cubes. The samples were demolded after 24 h, wrapped in a plastic sheet, and put back into the climatic chamber until testing. The compressive strength was tested at the desired ages (every 7 h from the time of casting until 70 h and 168 h). Two samples were tested at each age. 

## 3. Individual Test Results

### 3.1. Slump Flow

Figure 2 shows the slump flow evolution from various slag fly-ash binders. The first recorded flow diameters after 10–15 min amount to 28–45 cm, whereby the slump flow, as expected, was higher for mixes with high s/b ratios. The flow diameters decreased rapidly for all mixes, and the flow was lost (the flow diameter was equal to the diameter of the cylinder, amounting to 10 cm) after 45–60 min. Compositions with low s/b ratios reached the 10 cm threshold earlier than compositions with a high s/b ratio, except for the composition SF–070SB–HA. Its flow decreased rapidly, due to the high Na_2_O and SiO_2_ content (see Table 3). This observation was supported by previous experimental findings [59,64], where high Na_2_O dosage was found to promote the dissolution of slag at a very early age, in addition to increased SiO_4_^4-^ ion concentration that assisted in the rapid formation of a solid percolation path. Moreover, the time at which no virtual paste spreading was observed in the slump flow test was taken as a criterion for the initial setting time (t_sf_) for all the studied mixes. In other words, the (initial) setting time, obtained from the slump flow tests, was considered as the time when the flow diameter was equal to 10 cm, which was the diameter of the bottom opening of the cone.

### 3.2. Vicat Test

Figure 3 shows the influence of varying s/b ratios and alkali contents on setting times of slag fly-ash binders, determined by the Vicat needle test. The initial setting times (t_i_) ranged from 45 to 100 min while the final setting times (t_f_) recorded were between 80 and 175 min. By analogy to the slump flow, for compositions with the same alkali content (SF—050SB, 055SB, and 064SB), the setting times increased with an increasing s/b ratio. Lower solution quantities resulted in higher initial solid volume fractions, enabled a faster formation of a continuous network of reaction products, and, thus, resulted in a faster setting. However, for the mix with the lowest solution–binder ratio (SF—047SB—LA), initial and final setting times were comparable to SF—050SB, due to low alkali and silica content in the mix, which decelerated the very early age reaction. In the case of SF—070SB—HA mix, final setting times were comparable to SF—064SB, due to the presence of higher silica content in the activating solution that formed additional early-reaction products [65]. The initial setting time for SF—070SB—HA, however, was still higher than that of SF—064SB, revealing that this reaction boost due to additional alkalis did not compensate for the initial setting delay due to the higher s/b ratio.

### 3.3. Isotheral Calorimetry

The evolutions of the calorimetry-determined heat flow q (t) and the cumulative heat Q (t) are depicted in Figure 4. Notably, due to external mixing, only a part of the first peak was recorded. This initial peak was related to the rapid dissolution of the slag as soon as it came in contact with the alkaline activators [65]. The peak was followed, for all mixes, by a rapid decrease in the heat flow until a minimum was reached between 12 and 24 h. This, in turn, was followed by a second, very pronounced heat-flow peak, which was known to be a result of the polymerization of dissolved silicate and aluminate units [52,66]. There was a very noticeable shift of this main reaction peak, both in terms of age of occurrence and amplitude. For the mixes with the same alkali content, the blue lines in Figure 4a may be compared. The peak occurred earlier and was higher but less wide the lower the s/b ratio was. Interestingly, however, the three mixes exhibited virtually the same amount of cumulative heat of 120 J/g at the end of the monitoring at an age of 96 h. This indicated that the reaction for the mix with the lowest s/b ratio was already limited by the availability of the solution, allowing the others to catch up. The mix with the highest s/b ratio (SF—070SB—HA) showed a shorter dormant period than the mix SF—064SB, due to high alkali and silica contents that facilitated the formation of reaction products and manifested in higher cumulative heats (Figure 4b) at 96 h, compared to the other mixes. The main heat-flow peak for the low-alkali mix (SF—047SB—LA) occurred later than that of the mix SF—050SB, despite the lower s/b ratio, demonstrating that a shortage of alkali decelerated the hydration.

Next, we aimed to identify a setting time based on the heat-flow curves. However, all characteristic points (the start of the second acceleration, as well as the second peak) occurred well after the setting of the material. In more detail, an early-age minimum heat flow (equal to the start of the second acceleration) was reached at 12 h or even later, while the slump flow was already lost after roughly one hour (see Figure 2), and Vicat tests resulted in setting times of 45 minutes to two hours (Figure 3). In conclusion, isothermal calorimetry did not allow for extracting quantitative information on the setting.

As for the evaluation and discussion of the results, we defined a degree of reaction r as the ratio between the cumulated heat release Q(t) at any age and the ultimate heat release $Q_{\infty }$ at an infinite time [67,68]:(3)$$r\;\left(t\right)=\frac{Q\left(t\right)}{Q_{\infty }}$$

To estimate the ultimate heat release $Q_{\infty }$, the heat flow was extrapolated. Therefore, we followed [69] and plotted the cumulative heat Q as a function of the inverse square root of time and considered that Q linearly increased with decreasing the inverse square root of time—see Figure 5a for the extrapolation procedure for the mix SF—047SB—LA. This yielded the sought ultimate heat, $Q_{\infty }$, as the intercept with the vertical axis. This allowed us to plot cumulative heat (from the actual measurement and extrapolation) and the respective degree of reaction up to later ages—see Figure 5b for the evolution of mix SF—047SB—LA. Comparing the resulting ultimate heat of all five mixes (Table 4) shows that the mixes with increasing s/b ratio reached higher ultimate heat at an infinite time. This was due to higher solution availability in the system, resulting in a higher reaction degree at an infinite time. Mixes with high s/b ratios, SF—070SB—HA, achieved higher ultimate cumulative heats among all studied mixes at an infinite time, due to the higher availability of the activating solution with high alkali and silica contents that facilitated higher product formation over time. 

### 3.4. Ultrasonic Measurements

Figure 6 depicts the ultrasonic pulse velocity (UPV) evolution of slag-fly-ash binders for 3 days (72 h). To highlight the very-early-age velocity variations on the UPV curves, a logarithmic scale was used, as shown in Figure 7a. The initial P-wave velocity was around 190–250 m/s for all the studied mix compositions, which was lower than the velocity in the air (340 m/s) and alkaline solution (1450 m/s) [52]. This low UPV velocity was due to the air bubbles entrapped during solution and sample preparation [70]. Wetting and dissolution (stage 01) of slag and fly ash occurred in this stage, resulting in aluminosilicate monomers in solution [71]. All the slag-fly ash binder mixes followed a constant velocity until the end of the dissolution period, followed by a steep increase in P-wave velocity (stage 02—acceleration or condensation period). This sharp increase in velocity was attributed to the continuous condensation reaction and enhanced connected solid volume fraction. Among the three mixes, SF—050SB, SF—055SB, and SF—064SB, with the only difference being the s/b ratio, SF—050SB showed an earlier increase in velocity on a UPV curve, indicating a faster setting time. This implied that a lower solution quantity facilitated a quicker solid percolation path, allowing the formation of a continuous chain of hydration products and resulting in a denser microstructure to better propagate the P-wave. 

For the SF—047—LA mix, even though s/b was the lowest among all of the mixes, corresponding transitions on the UPV curve were comparable to the SF—050SB mix. One of the possible reasons for this effect may be the reduced alkali content that slows down the alkali activation in the mix [72]. In contrast, the sample with the highest solution-to-binder ratio and high alkali content, SF—070SB—HA, demonstrated a faster transition of P-wave velocity in the acceleration period on the UPV curve, compared to SF—064SB. This was mainly due to increased silica and alkali content in the mix that assisted in improving reaction products at an early age. Moreover, a higher Na_2_O dosage increased the alkalinity of the mix, promoting a higher dissolution of slag [73]. An induction period (stage 03) was noticed for all mixes as the P-wave velocity increased slowly followed by a secondary acceleration period (stage 04), due to the gradually increasing total solid volume fraction.

For all five mixes, the UPV evolved in a characteristic double S-shaped fashion, motivating a mathematical treatment to analyze the measurements and compare the different mixes. The measured UPV evolution was fitted by means of superimposing three logistic functions, reading, as in [29]:(4)$$V\left(t\right)=\overset{3}{\underset{i=1}{\sum }}\left(\frac{V_{i}}{1+e^{\left(t-t_{i}\right)/g_{i}}}\right)+c$$
with gradients $g_{i}$, inflection points $t_{i}$, plateaus or asymptotic values $V_{i}$, and an additive constant c. Table 5 shows the results in root mean square error (RMSE) for all studied mixes. Detailed information on the fitting procedure can be found in [74]. The quality of the fit was excellent; RMSE was lower than 30 m/s, which was demonstrated to be exemplary for mix SF—047SB—LA (see Figure 7b).

The (modeled) UPV curve allows for identifying characteristic stages, as described next. Therefore, three straight lines were introduced (see Figure 8). The first one was fitted to the very first UPV measurements. The second line was a tangent through the first inflection point (IP_1_) at $t_{1}$. The third straight line, in turn, was a tangent through the second inflection point (IP_2_) at $t_{2}$. The intersection of the first and second regression lines marked the transition of the dissolution period (stage 01) to the acceleration or condensation period (stage 02) and the transition point was a candidate for the ultrasound-derived initial setting time [35,57], labeled t_Vp(int-1)_. Another candidate for the beginning of setting or initial setting was the time at the first inflection point itself [75,76], labeled t_Vp’(max)_. The intersection of the second and third straight line marked the transition to the dormant or induction period (stage 03), and was considered a candidate for the final setting time [35,57] and labeled t_Vp(int-2)_. Alternatively, the final setting was assigned to the time t_0.2*Vp’(max)_ when the derivative of the P-wave velocity dropped down to 20% of its maximum [34]. This was followed by the so-called secondary acceleration period (stage 04), which ended at the third inflection point (IP_3_). Some studies suggested an additional stage (stage 05—stable stage), where the P-wave velocity gradually increases until it reaches an asymptotic value [34,55]. In this particular study, this stage was not found within the 72 h-long measurements. Derivative method and intersection method are two terms that are used hereafter for derivative-based and intersection-based identification of characteristic stages, linked to the setting time, on a UPV curve. Initial and final setting times obtained through the different criteria are listed in Table 6.

As for a final evaluation of the UPV evolutions, we aimed at discussing the eligibility of the frequently adopted velocity thresholds [28,29,36,49,51,52,53] (see also Table 1) to quantify the initial and final settings. Figure 9 depicts and Table 7 lists the P-wave velocities at the setting times obtained from slump flow, Vicat test, and the characteristic points from the UPV tests, respectively, for all the slag-fly ash binder mixes. Interestingly, the initial setting times from the Vicat and slump flow tests corresponded to a wide range of P-wave velocities within the interval from 411 to 1847 m/s. This showed that intrinsic velocity thresholds that correspond to the initial setting could not be defined for our different mixes. By analogy, the P-wave velocities that corresponded to the final setting from Vicat tests were not at all constant for all mixes, making the definition of a threshold impossible. Even the measured P-wave velocities that corresponded to the UPV-determined characteristic points were not constant for the different mixes, particularly not for the mixes with different alkalinities.

### 3.5. Compressive Strength Evolution

Figure 10 shows the compressive strength development of slag-fly ash binders. The samples were tested every 7 h from the time of casting, up to 70 h, and additionally at 168 h (7 days). The largest compressive strength was observed, at any age, for the mix SF—050SB. At 7 days, the mix exhibited a strength of roughly 67 MPa, thus clearly outperforming the other mixes. As expected, the mixes with a larger s/b ratio exhibited a lower strength. Interestingly, the strength of the SF—064SB mix remained below 5 MPa, even after an age of 48 h. This demonstrated that the higher water content in the mix implied a weak paste at an early age. On the contrary, a lower w/b ratio resulted in a higher solid volume fraction, and the formation of more reaction products that link together to produce a dense microstructure, which, in turn, resulted in a faster increase of the compressive strength over time. In the case of SF—047SB—LA, even though the s/b ratio was the lowest among all the mixes, the strength results were lower than those of SF—050SB and comparable to those of SF—055SB. This observation can be explained by the smaller alkali content in the mix that resulted in decelerated slag dissolution at a very early age and, thus, fewer reaction products (at the same age) compared to those of SF—050SB. The mix with the highest s/b ratio, SF—070SB—HA, showed comparable strengths to SF—055SB at an early age, due to the presence of high alkali content. Higher silica content in the activating solution promoted Si-rich C-A-S-H precipitation [48,77] and, hence, an increase in compressive strength. 

Next, mechanical percolation thresholds (MPT) were defined. Therefore, the measured strength evolutions were studied as a function of the calorimetry-based evolution of the degree of reaction (see Figure 11). To quantify the MPTs, two different functions were fitted to the depicted strength ($f_{c}$) vs. degree of reaction (α) evolutions, as was done in earlier works [69] and [78]:A linear function of the form
(5)$$f_{c}\left(\alpha \right)=\left(\alpha -\alpha_{\mathrm{lf}}\right)\cdot k$$
with two fitting parameters $\alpha_{\mathrm{lf}}$ denoting the degree of reaction at the mechanical percolation threshold and k as slope (see Table 8 for numerical values). The resulting straight-line fits represented the observed strength evolution fairly well (Figure 11a), as quantified by average RMSE, amounting to 2.31 MPa.

Alternatively, a power function of the form(6)$$f_{c}\left(\alpha \right)=f_{c,\infty }{\left(\frac{\alpha -{\;\alpha }_{\mathrm{pf}}}{\alpha_{\infty }}\right)}^{\beta }$$ was fitted to the experimental data, where $f_{c,\infty }$ is the compressive strength for a reaction degree of $\alpha_{\infty }=0.75$, $\alpha_{\mathrm{pf}}\;$ denotes the degree of reaction at the mechanical percolation threshold, and β is the power law exponent. Numerical values of the three fitting parameters, $f_{c,\infty }$, $\alpha_{\mathrm{pf}}\;,\;$ and β, along with RMSE, are listed in Table 8. It is also important to note that RMSE was generally smaller with the power law fit, compared to the linear fit for all studied mixes; the average RMSE amounted to 1.35 MPa. This indicated the quality of the fit, using this model, to link the strength evolution to the degree of reaction of all the studied mix compositions. 

Figure 11a showed a linear fit between compressive strength evolution and corresponding reaction degrees of all slag-fly ash binders. It was evident that reaction degrees were overestimated in this case for mechanical percolation (MPT_lf_) to occur, as the compressive strength values showed a significant development already. On the other hand, for a power-law fit; to determine the mechanical percolation threshold (MPT_pf_), Figure 11b, compressive strength evolutions are in good agreement with the corresponding degrees of reaction of all studied mixes except SF—064SB. The power-law fit for the SF—064SB mix showed a mechanical percolation (MPT_pf_) to occur at the reaction degree of ${\;\alpha }_{\mathrm{pf}}=0$. This value is unrealistic as it implies already a strength gain when the first ions dissolve and generate heat. However, non-zero strength requires the precipitation of enough reaction products. The degrees of reaction at the mechanical percolation threshold, criteria obtained by a linear function and power function, are listed in Table 8. The power-law fit results are consistent with other experimental studies [79,80] which at the final setting and compressive strengths amounting to roughly 0.50 MPa [81]. These low degrees of reaction at the final set can be attributed to the very early age slag dissolution and rapid reaction product formation (primary C-S-H formation by dominant silicate ions) around the unreacted slag-fly ash particles that hinder the continuation of reaction (pre-dormant period on heat flow curves) until the critical concentration of dissolved units was achieved to produce further reaction products (post-dormant period) [68].

In contrast, previous studies on OPC-based concrete with s/b = 0.5, identified reaction degrees of 0.075–0.13 for initial and final setting, respectively [50,82]. These significant differences can be attributed to two phenomena: firstly, to different scales of testing (paste, mortar, or concrete mixtures), and secondly, to different ultimate heats $Q_{\infty }$. In our current study, the value was approximated through extrapolation of the experimental heat flow data at an infinite time, while those previously mentioned studies opted for the maximum cumulative heat measured at 7 days to calculate the reaction degrees.

Furthermore, all samples achieved a considerable compressive strength before the dormant period. We conclude that setting occurred in the pre-dormant stage. Interestingly, these results are in contrast to findings of Bernal et.al [66] and Yang et.al [72] who determined the final setting in the acceleration stage (similar to conventional Portland cement [20]). The former study used slag-metakaolin blends (secondary peak on heat flow curves obtained within 2 h) while the latter utilized slag-fly ash along with the varying quantities of fly ash microspheres (acceleration period achieved within 1–2 h) in their study. 

## 4. Discussion

Herein, the setting times, as identified by means of the different tests, are compared in detail (see Figure 12 for a schematic overview of the test results). In the case of the first three mixtures, samples with the same alkali content but varying s/b ratio, both the initial and final setting times increased with the increasing s/b ratio (in all the studied criteria) due to the increasing distance between the precursor particles in the fluid mix. The low alkali mix SF—047SB—LA consistently showed the earliest setting out of all five, given that it exhibited the lowest s/b ratio. The setting of the high alkali mix SF—070SB—HA was very close to the setting of the SF—064SB mix; depending on the test technique, it sometimes may even have occurred earlier, despite the larger s/b ratio. This underlines that the high content of alkali accelerated the formation of reaction products, which was particularly visible when comparing the setting durations, i.e., the difference between the final and initial setting times. This difference was much smaller for the SF—070SB—HA, compared with that of SF—064SB. 

Next, the setting times for the different techniques were compared. For the two mixes with SF—047SB—LA and SF—050SB, the initial and final setting times determined from different criteria were in excellent agreement. The initial setting times from the UPV and Vicat tests were particularly close to the slump loss, which indicated that either test yields trustworthy setting results for low s/b mixes.

However, for mixes with a higher s/b ratio, SF—055SB, SF—064SB and SF—070SB—HA, the different techniques yielded rather different results. The Vicat tests indicated that initial setting occurred much later, while the slump flow and UPV techniques, particularly the intersection method results, were relatively close. This could be attributed to the fact that the loss of the flowability in a slump test and the first intersection point of the ultrasonic pulse velocity test were both sensitive to the first solid percolation path, which occurred at the beginning of the acceleration period (Stage 02 in Figure 8) [83]. Even though the precursor grains connected, the connectivity or the cohesion was not enough to indicate the beginning of a mechanical set [84,85]. Thus, we concluded that setting determination in high s/b systems is more accurate with slump flow and UPV tests, while the sensitivity of Vicat tests is not sufficiently high. The non-destructive UPV tests were particularly interesting, because they were sensitive to changes in the microstructure of the hardening mix. For example, in the case of SF—070SB—HA, the presence of high alkali and silica content in the activating solution promoted Si-rich reaction products at a very early age, and this microstructure change was tracked by the earlier occurrence of the first inflection point on the UPV curve (see Table 6 and Figure 12). 

The mechanical percolation threshold criterion obtained through linear as well as power/law fitting overestimated the time of occurrence of a mechanical set, compared to the time obtained via other criteria. This showed that any extrapolation attempts from destructive strength testing at early ages fails to provide accurate information on setting.

## 5. Conclusions and Outlook

This study focused on investigating the capability of five different destructive and non-destructive measurements on the determination of the setting behavior of alkali-activated slag-fly ash binders, prepared with various amounts of alkali, silica, and s/b ratio. The main conclusions drawn from this study are as follows:The ultrasonic pulse velocity (UPV) measurements for all slag-fly ash binders revealed four characteristic stages of velocity evolutions, clearly sensitive to the alkali and silica dosages as well as to the s/b ratio.The UPV-derived initial setting times showed good agreement with the slump flow measurements for all the studied mixes. Interestingly, the classical Vicat tests were accurate for mixes with low s/b ratios of 0.5 or smaller; for increasing s/b ratios, Vicat initial setting times occurred significantly later than UPV-derived initial setting times, and then the slump loss.P-wave velocities at the identified (initial and final) setting times are very different from one mix to another. Thus, P-wave velocity thresholds cannot be used for setting quantification of the studied AASF mixes.Heat flow evolutions obtained from isothermal calorimetry could not be used to determine setting times, as the characteristic minimum and maximum heat flow occurred at later ages (dormant period, acceleration stage, and appearance of main hydration peak).For mixes with the same alkali and silica content, the compressive strength decreased with an increasing solution-to-binder (s/b) ratio. This trend was expected, as a lower s/b ratio facilitates the closer packing of precursor particles, fewer pores, and, thus, a stronger microstructure. The compressive strength increased progressively with increasing calorimetry-derived reaction degrees. The correlation could be fitted reasonably well using a power-law function, but extrapolation attempts to predict the setting were not accurate.

As a perspective for characterizing the setting and hardening behavior of the mixes, a Temperature Stress Testing Machine (TSTM) can be used. This allows for the continuous monitoring of the evolution of elastic stiffness since the earliest age [26,86]. This way, the onset of mechanical percolation can be determined and linked to the final setting time determined herein, based on indirect techniques.

## 6. Limitation of the Study

This study is limited to slag-fly ash paste activated by sodium hydroxide and sodium silicate at a curing temperature of 20 °C.

## Figures and Tables

**Figure 1 materials-16-00373-f001:**
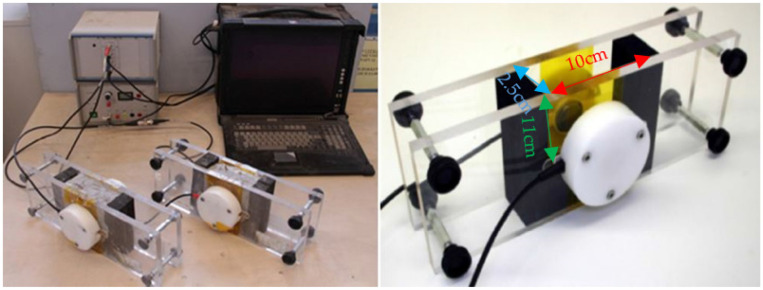
FreshCON system [63] for measuring the ultrasonic pulse velocity.

**Figure 2 materials-16-00373-f002:**
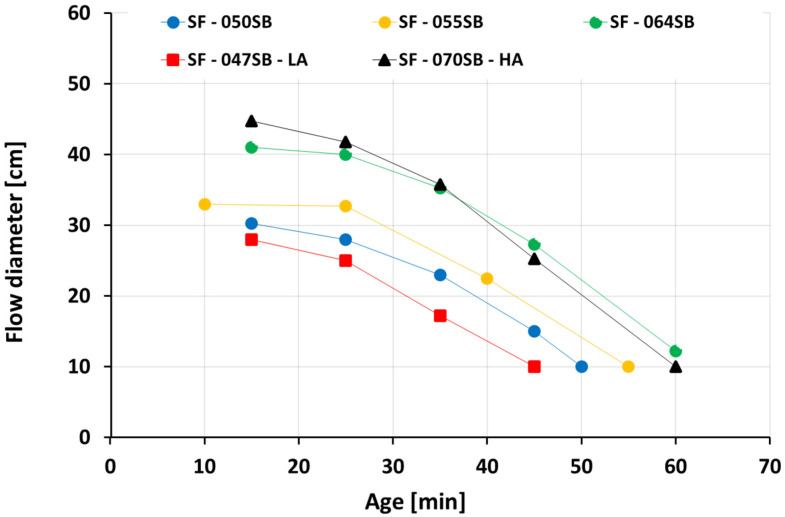
Slump flow evolution as a function of age.

**Figure 3 materials-16-00373-f003:**
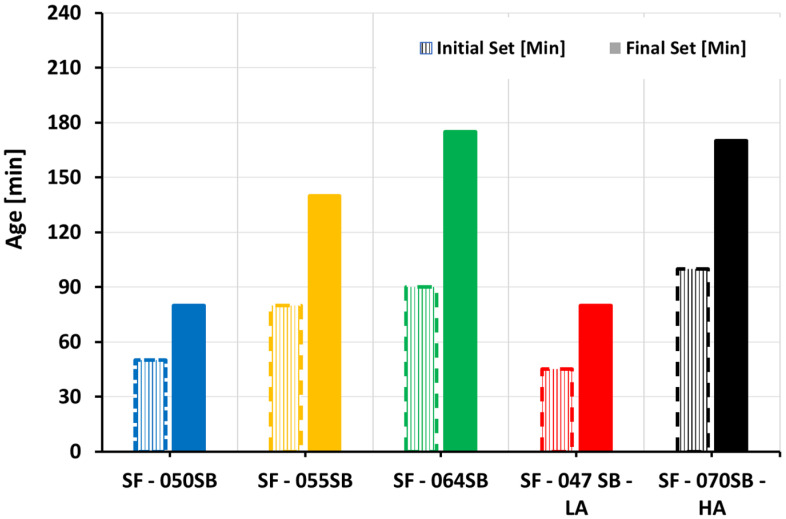
Initial and final setting times obtained from Vicat tests.

**Figure 4 materials-16-00373-f004:**
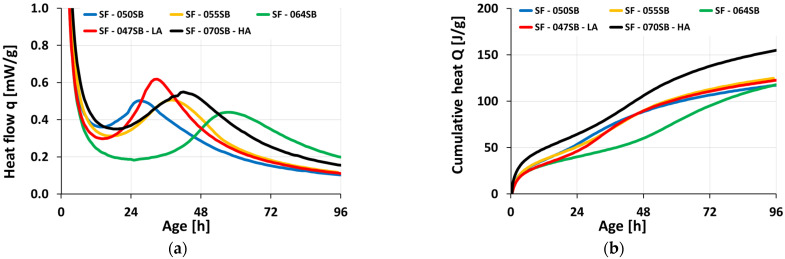
Isothermal calorimetry as a function of age: (**a**) heat flow; (**b**) cumulative heat release.

**Figure 5 materials-16-00373-f005:**
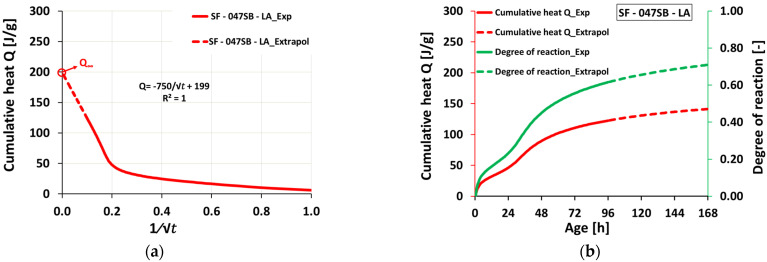
Extrapolation procedure for determination of the ultimate heat after [68] for mix SF—047SB—LA: (**a**) cumulative heat as a function of 1/√t; (**b**) evolution of cumulative heat, as well as the respective degree of reaction, including the actual measurements.

**Figure 6 materials-16-00373-f006:**
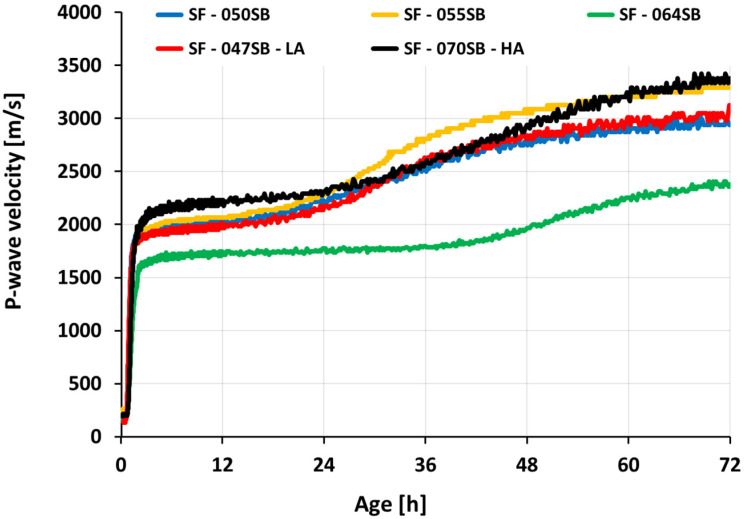
UPV evolution as a function of age.

**Figure 7 materials-16-00373-f007:**
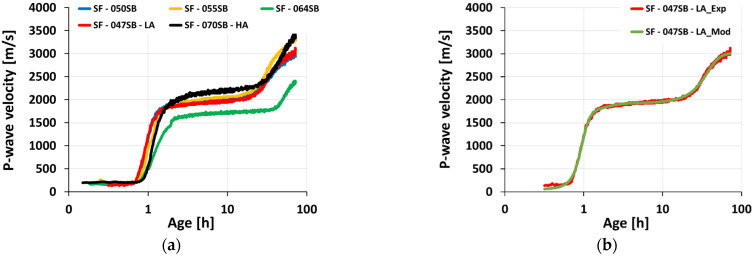
UPV evolution as a function of age: (**a**) all studied mixes in logarithmic scale; (**b**) reference modeled UPV curve for mix SF—047SB—LA.

**Figure 8 materials-16-00373-f008:**
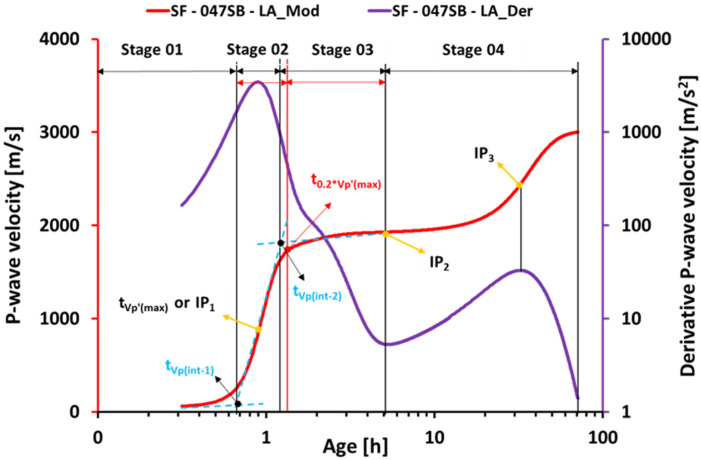
Different stages, intersections, and inflection points on the UPV curve as a function of age.

**Figure 9 materials-16-00373-f009:**
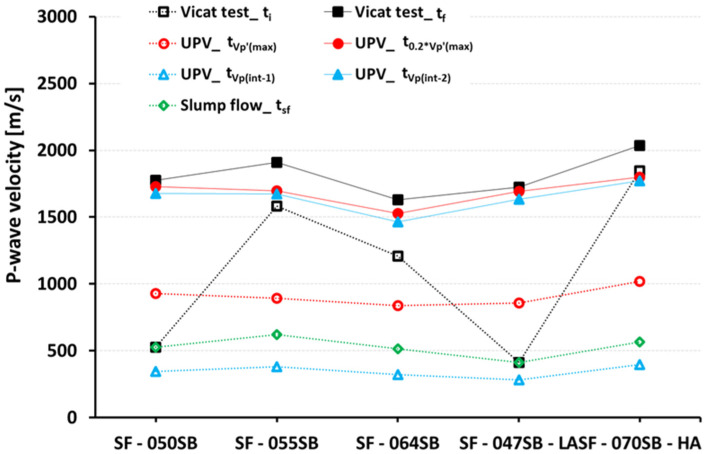
P-wave velocities that correspond to the setting times determined by slump flow tests, Vicat tests, and UPV tests, plotted for all five mixes. Hollow markers with dashed lines depict the initial setting, while solid markers with solid lines show the final setting.

**Figure 10 materials-16-00373-f010:**
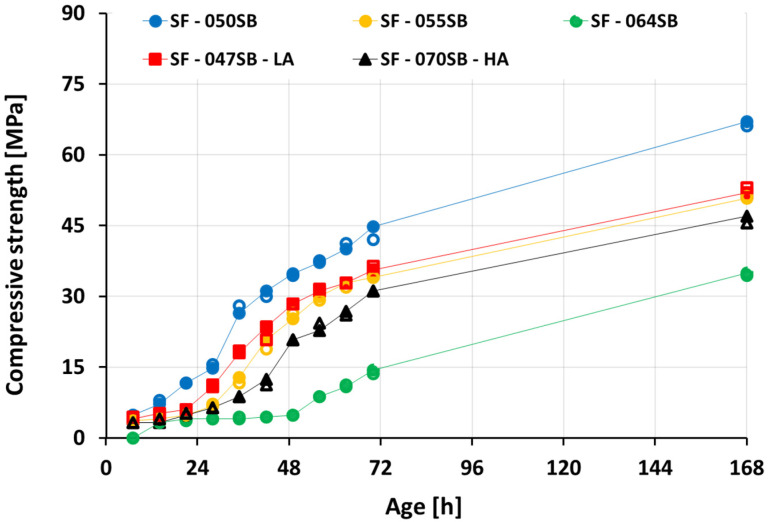
Compressive strength development as a function of age. All strength tests were performed on two samples—compare the hollow and solid markers for the individual results.

**Figure 11 materials-16-00373-f011:**
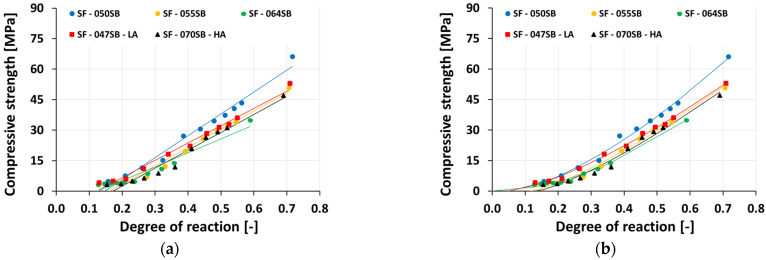
Compressive strength evolution as a function of the degrees of reaction: (**a**) linear fit; (**b**) power-law fit.

**Figure 12 materials-16-00373-f012:**
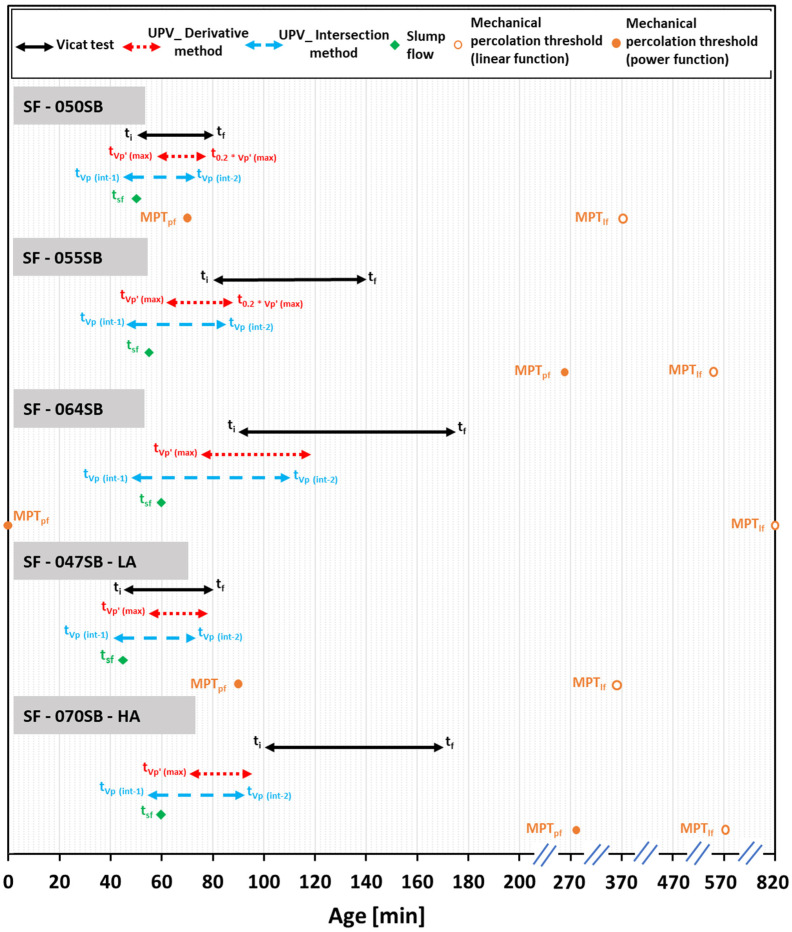
Timeline for Initial and final setting time values as obtained through various techniques.

**Table 2 materials-16-00373-t002:** Oxide compositions of precursors (wt.%) [58].

Oxide Composition	Slag [%]	Fly Ash [%]
CaO	40.80	4.33
SiO_2_	33.30	56.70
Al_2_O_3_	12.30	23.50
MgO	7.84	1.43
SO_3_	2.30	1.16
TiO_2_	1.29	1.23
K_2_O	0.67	2.65
Na_2_O	0.44	0.91
Fe_2_O_3_	0.39	5.92
MnO	0.36	-
BaO	0.11	0.21
SrO	-	0.15
P_2_O_5_	-	1.49

**Table 3 materials-16-00373-t003:** Mix design for alkali-activated slag-fly ash binders (100 g of binder).

Parameters	SF—050SB	SF—055SB	SF—064SB	SF—047SB—LA	SF—070SB—HA
Slag [g]	50	50	50	50	50
Fly ash [g]	50	50	50	50	50
NaOH (8 M aq) [g]	2.49	2.49	2.49	2.13	3.11
Na_2_SiO_3_ (aq) [g]	26.53	26.53	26.53	22.91	33.79
Water [g]	20.85	25.85	34.85	22.07	32.47
s/b [-]	0.50	0.55	0.64	0.47	0.70
w/b [-]	0.37	0.42	0.51	0.36	0.53
Na_2_O content [%]	5.26	5.26	5.26	4.54	6.69
SiO_2_ content [%]	7.56	7.56	7.56	6.53	9.63
Ms value [-]	1.44	1.44	1.44	1.44	1.44

**Table 4 materials-16-00373-t004:** Estimated ultimate heat release $Q_{\infty }\;$ through extrapolation of calorimetry results.

Ultimate Heat Release	SF—050SB	SF—055SB	SF—064SB	SF—047SB—LA	SF—070SB—HA
$Q_{\infty }\;\left[J/g\right]$	187.6	204.7	258.8	199.2	262.2

**Table 5 materials-16-00373-t005:** RMSE of the modeled curves of UPV evolution (P-wave) for all mix compositions.

	SF—050SB	SF—055SB	SF—064SB	SF—047SB—LA	SF—070SB—HA
RMSE [m/s]	19.83	27.85	18.88	25.46	26.33

**Table 6 materials-16-00373-t006:** Initial and final setting times obtained from different techniques.

	Initial Setting Time [min]				Final Setting Time [min]		
Mix ID	Slump Flow [t_sf_]	Vicat Test [t_i_]	FreshCON [t_Vp’(max)_]	FreshCON [t_Vp(int-1)_]	Vicat Test [t_f_]	FreshCON [t_0.2*Vp’(max)_]	FreshCON [t_Vp(int-2)_]
SF—050SB	50	50	58	45	80	77	73
SF—055SB	55	80	62	46	140	88	85
SF—064SB	60	90	75	48	175	118	110
SF—047SB—LA	45	45	55	41	80	78	73
SF—070SB—HA	60	100	71	54	170	95	92

**Table 7 materials-16-00373-t007:** Summary P-wave velocity that corresponds to the setting times determined by slump flow tests, Vicat tests, and UPV tests, plotted for all five mixes [m/s].

Mix ID	UPV Values at Initial Set [m/s]				UPV Values at Final Set [m/s]		
	Slump Flow [t_sf_]	Vicat Test [t_i_]	FreshCON [t_Vp’(max)_]	FreshCON [t_Vp(int-1)_]	Vicat Test [t_f_]	FreshCON [t_0.2*Vp’(max)_]	FreshCON [t_Vp(int-2)_]
SF—050SB	525	525	928	344	1775	1730	1678
SF—055SB	620	1581	892	380	1910	1696	1675
SF—064SB	516	1209	840	322	1631	1526	1466
SF—047SB—LA	411	411	857	282	1725	1694	1635
SF—070SB—HA	565	1847	1020	397	2037	1798	1771

**Table 8 materials-16-00373-t008:** Parameters of the fitting function, Equations (5) and (6), for the compressive strength evolution as a function of the degree of reaction.

**Mix ID**	Linear Function, MPT_lf_				Power Function, MPT_pf_				
	Slope k	RMSE	$\mathrm{Reaction}\;\mathrm{Degrees}\;\alpha_{lf}$	Age	Compressive Strength $f_{c,\infty }$	Power Law Exponent β	RMSE	$\mathrm{Reaction}\;\mathrm{Degrees}\;\alpha_{Pf}$	Age
	[MPa]	[MPa]	[-]	[min]	[MPa]	[-]	[MPa]	[-]	[min]
SF—050SB	64.89	2.53	0.147	370	70.31	1.46	1.48	0.052	70
SF—055SB	52.81	2.28	0.161	530	56.94	1.30	1.51	0.118	260
SF—064SB	42.90	2.32	0.126	820	54.72	1.86	0.85	0.000	00
SF—047SB—LA	53.46	1.81	0.122	360	57.59	1.38	0.93	0.050	90
SF—070SB—HA	50.97	2.62	0.172	570	55.50	1.30	1.20	0.132	280

## Data Availability

Not applicable.

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
