# Peer review of "Effect of Solution-to-Binder Ratio and Alkalinity on Setting and Early-Age Properties of Alkali-Activated Slag-Fly Ash Binders"

_materials, 2022, doi:10.3390/ma16010373_

Round 1

Reviewer 1 Report

·        The study is original and interesting. Especially fresh properties of the alkali activated blended binders have been evaluated. Some amendments are required before publication:

·         The densities of fly ash and slag should be given. The purity of NaOH should be presented.

·         “The concentration of NaOH solution was 8M” why this concentration was selected? Is there a reference or a preliminary work?

·         Similarly for Ms value and w/b ratios..

·         Additional published works about the high-Ca alkali-activated systems should be mentioned and results can be compared with these works: “Effects of Ms modulus, Na concentration and fly ash content on properties of vapour-cured geopolymer mortars exposed to high temperatures” and “Investigation of fire and chemical effects on the properties of alkali-activated lightweight concretes produced with basaltic pumice aggregate”

·         Discussions were conducted in a procedural and detailed manner.

Author Response

Authors’ Response to the Reviewer Comments on
“Effect of solution-to-binder ratio and alkalinity on setting and early-age properties of alkali-activated slag-fly ash binders” by Ali et al.

We have revised our manuscript in accordance with the reviewer comments to improve the manuscript. We are sincerely grateful to the reviewer for his elaborated comments. We respond to each review comment or suggestion made by the reviewer in detail below.

The study is original and interesting. Especially fresh properties of the alkali activated blended binders have been evaluated. Some amendments are required before publication:

  1. The densities of fly ash and slag should be given. The purity of NaOH should be presented

 In accordance with the reviewer’s comment, the relevant information has been added in the revised manuscript as follows.

“The densities of GGBFS and FA are 2.92 and 2.32 (g/cm3), respectively.” Sodium hydroxide (NaOH) with 97% purity, in powdered form…”

(107th thru 108th line on Page 3 of revision)

  1. The concentration of NaOH solution was 8M” why this concentration was selected? Is there a reference or a preliminary work? Similarly, for Ms value and w/b ratios.

Concentration of NaOH:

The 8M concentration of NaOH solution does not have any significance. In literature, the added NaOH concentration varies between 4M to 10M to achieve a desired Ms [1,2] . If we would have used a lower NaOH concentration, instead of 8M, then there would be lower amount of water. Furthermore, going up to 10/12M NaOH, the reaction kinetics gets slower.

Ms and w/b:

It was reported in literature that the highest mechanical and durability properties can be achieved with a solution having Ms value in range of 1.0 – 1.5 [3]. Thus, in this study, all mix compositions were prepared at a fixed alkali modulus (Ms = 1.44). For w/b and s/b, initially we tried to reduce/increase those of the same amount, but some mixes were way too dry to handle; that’s why the reduction in those parameters is lower than their increase.

  1. Additional published works about the high-Ca alkali-activated systems should be mentioned and results can be compared with these works: “Effects of Ms modulus, Na concentration and fly ash content on properties of vapour-cured geopolymer mortars exposed to high temperatures” and “Investigation of fire and chemical effects on the properties of alkali-activated lightweight concretes produced with basaltic pumice aggregate

To reflect the reviewer’s comments, the references have been newly added in the revised manuscript as follows.

“Blending slag and fly ash, together with the choice of suitable activators, overcomes the aforementioned drawbacks and provides both somewhat contradicting requirements for modern AAMs: workable mixes and high early-age strengths [16-19] [19]4.”

(42nd thru 45th line on Page 1 of revision)

“Moreover, a higher Na2O dosage increases the alkalinity of the mix, promoting a higher dissolution of slag [73]5.

(317th thru 318th line on Page 9 of revision)

[1] Joseph, S.; Cizer, Ö. Comparative Analysis of Heat Release, Bound Water Content and Compressive Strength of Alkali-Activated Slag-Fly Ash. Front. Mater. 2022, 9, 1–11,

2 Chithiraputhiran, S.; Neithalath, N. Isothermal Reaction Kinetics and Temperature Dependence of Alkali Activation of Slag, Fly Ash, and Their Blends. Constr. Build. Mater. 2013, 45, 233–242,

3 Provis, J.L.; Kilcullen, A.; Duxson, P.; Brice, D.G.; Van Deventer, J.S.J. Stabilization of Low-Modulus Sodium Silicate Solutions by Alkali Substitution. Ind. Eng. Chem. Res. 2012,

4 Sarıdemir, M.; Çelikten, S. Investigation of Fire and Chemical Effects on the Properties of Alkali-Activated Lightweight Concretes Produced with Basaltic Pumice Aggregate. Constr. Build. Mater. 2020

5 Sarıdemir, M.; Çelikten, S. Effects of Ms Modulus, Na Concentration and Fly Ash Content on Properties of Vapour-Cured Geopolymer Mortars Exposed to High Temperatures. Constr. Build. Mater. 2023,

Reviewer 2 Report

1.       A lot of disjointed sentences makes the flow of the article difficult for the reader. Abstract and the entire article should be proofread by an expert in the field

2. The loss of workability, determined from the time when the slump flow becomes negligible, correlates well with characteristic points on the ultrasonic P-wave velocity evolutions doesn’t  but this was not the same for Vicat or calorimetry test results of all mixes. See (Line 22)

3.       The word behaviour was spelt as “behavior” in more than two instances. Kindly locate this and change appropriately. Also, there is an intermixing in the use of UK English and USA English. Author should stick to one for uniformity.

4.       In this study, we investigate the efficacy of both ultrasonic and calorimetric tests to decipher the setting of alkali-activated slag fly-ash binders was investigated. (See line 88)

5.       To monitor a challenging task such as the development of the mechanical properties of fragile samples near or after setting, uniaxial compressive strength tests were performed. (See Line 91-92)

6.       The flow diameters decreased rapidly for all mixes, and the flow was lost (See line 202 -203)

7.       Author to use different colours to represent SF-055SB and SF-064SB in Figures 2,3, 4 (a and b), 6, 7a, 10 and 11. At a glance, they look too similar to SF-050SB.

8.       The author should try to point out the academic novelty/relevance of the results obtained from Isothermal calorimetry and UPV measurements.

9.       The word “precursor” as it has been used often in this context may not be a true representation of the meaning that the author wants. The author is advised to use other words that fit properly into the context. 

Overall manuscript needs some corrections to improve the reader interest. 

Author Response

Authors’ Response to the Reviewer Comments on
“Effect of solution-to-binder ratio and alkalinity on setting and early-age properties of alkali-activated slag-fly ash binders” by Ali et al.

We have revised our manuscript in accordance with the reviewer comments to improve the manuscript. We are sincerely grateful to the reviewer for his elaborated comments. We respond to each review comment or suggestion made by the reviewer in detail below.

  1. A lot of disjointed sentences makes the flow of the article difficult for the reader. Abstract and the entire article should be proofread by an expert in the field

 To reflect the reviewer’s comment, the entire article has been revised. We have ensured that the readability is improved, and that the individual abstracts are better connected.

  1. The loss of workability, determined from the time when the slump flow becomes negligible, correlates well with characteristic points on the ultrasonic P-wave velocity evolutions doesn’t but this was not the same for Vicat or calorimetry test results of all mixes. See (Line 22)

In accordance with the reviewer’s comment, the relevant sentence has been revised in the revised manuscript as follows.

“The loss of workability, determined from the time when the slump flow becomes negligible, correlates well with characteristic points on the ultrasonic P-wave velocity evolutions. This is, however, not the case for Vicat or calorimetry tests results for all mixes.

(21st thru 22nd line on Page 1 of revision)

  1. The word behaviour was spelt as “behavior” in more than two instances. Kindly locate this and change appropriately. Also, there is an intermixing in the use of UK English and USA English. Author should stick to one for uniformity.

In accordance with the reviewer’s comment, the relevant word has been revised in the revised manuscript.

  1. In this study, we investigate the efficacy of both ultrasonic and calorimetric tests to decipher the setting of alkali-activated slag fly-ash binders was investigated. (See line 88)

In accordance with the reviewer’s comment, the relevant sentence has been revised in the revised manuscript as follows.

“In this study, we investigate the efficacy of both ultrasonic and calorimetric tests to decipher the setting of alkali-activated slag fly-ash binders was investigated.”

(88th thru 89th line on Page 2 of revision)

  1. To monitor a challenging task such as the development of the mechanical properties of fragile samples near or after setting, uniaxial compressive strength tests were performed. (See Line 91-92)

To reflect the reviewer’s comment, the relevant sentence has been revised in the revised manuscript as follows.

“To monitor the development of mechanical properties of the fragile samples at early ages, near or after close to the setting, a challenging task, given the fragile nature of the samples at early age, uniaxial compressive strength tests are were performed with utmost care to not damage the samples before testing.”

(91st thru 94th line on Page 2 of revision)

  1. The flow diameters decreased rapidly for all mixes, and the flow was lost (See line 202 -203)

In accordance with the reviewer’s comment, the relevant sentence has been revised in the revised manuscript as follows.

“The flow diameters decreased rapidly for all mixes, and the flow is was lost.”

(202nd thru 203rd line on Page 6 of revision)

  1. Author to use different colours to represent SF-055SB and SF-064SB in Figures 2,3, 4 (a and b), 6, 7a, 10 and 11. At a glance, they look too similar to SF-050SB.)

In accordance with the reviewer’s comment, the relevant figures have been revised in the revised manuscript.

  1. The author should try to point out the academic novelty/relevance of the results obtained from Isothermal calorimetry and UPV measurements.

To reflect the reviewer’s comment, the relevant sentences have been newly added in the revised manuscript as follows.

“Next, we aim at identifying a setting time based on the heat flow curves. However, all characteristic points (the start of the second acceleration, as well as the second peak) occur well after the setting of the material…………”

(261st thru 284th line on Page 8 of revision)

Nonetheless In contrast, some previous studies on OPC-based concrete mixtures with s/b=0.5, suggested obtaining identified reaction degrees of 0.075 - 0.13 for initial and final setting………….”

(456th thru 470th line on Page 15 of revision)

“Next, the setting times for the different techniques are compared………”

(484th thru 506th line on Page 16-17 of revision)

“The UPV-derived initial setting times showed good correspondence agreement with to the slump flow measurements for all the studied mixes……………”

(520th thru 531st line on Page 17 of revision)

  1. The word “precursor” as it has been used often in this context may not be a true representation of the meaning that the author wants. The author is advised to use other words that fit properly into the context.

The word “precursor” had been frequently used in the literature to define the prime materials (e.g., slag and fly ash – calcium silicate or alumina silicate-rich solid precursors), used in the alkali-activated systems. However, to clarify the context of the word, as mentioned by the reviewer, the relevant word and associated text has been revised in the revised manuscript.
